# Anomaly Identification during Polymerase Chain Reaction for Detecting SARS-CoV-2 Using Artificial Intelligence Trained from Simulated Data

**DOI:** 10.3390/molecules26010020

**Published:** 2020-12-23

**Authors:** Reynaldo Villarreal-González, Antonio J. Acosta-Hoyos, Jaime A. Garzon-Ochoa, Nataly J. Galán-Freyle, Paola Amar-Sepúlveda, Leonardo C. Pacheco-Londoño

**Affiliations:** 1MacondoLab, Universidad Simón Bolívar, Barranquilla 080002, Colombia; reynaldovilla@gmail.com (R.V.-G.); jaime.garzon@unisimon.edu.co (J.A.G.-O.); nataly.galan@unisimonbolivar.edu.co (N.J.G.-F.); paola.amar@gmail.com (P.A.-S.); 2School of Basic and Biomedical Science, Universidad Simón Bolívar, Barranquilla 080002, Colombia

**Keywords:** SARS-CoV-2, artificial intelligence, polymerase chain reaction, COVID-19, simulated data

## Abstract

Real-time reverse transcription (RT) PCR is the gold standard for detecting Severe Acute Respiratory Syndrome Coronavirus 2 (SARS-CoV-2), owing to its sensitivity and specificity, thereby meeting the demand for the rising number of cases. The scarcity of trained molecular biologists for analyzing PCR results makes data verification a challenge. Artificial intelligence (AI) was designed to ease verification, by detecting atypical profiles in PCR curves caused by contamination or artifacts. Four classes of simulated real-time RT-PCR curves were generated, namely, positive, early, no, and abnormal amplifications. Machine learning (ML) models were generated and tested using small amounts of data from each class. The best model was used for classifying the big data obtained by the Virology Laboratory of Simon Bolivar University from real-time RT-PCR curves for SARS-CoV-2, and the model was retrained and implemented in a software that correlated patient data with test and AI diagnoses. The best strategy for AI included a binary classification model, which was generated from simulated data, where data analyzed by the first model were classified as either positive or negative and abnormal. To differentiate between negative and abnormal, the data were reevaluated using the second model. In the first model, the data required preanalysis through a combination of prepossessing. The early amplification class was eliminated from the models because the numbers of cases in big data was negligible. ML models can be created from simulated data using minimum available information. During analysis, changes or variations can be incorporated by generating simulated data, avoiding the incorporation of large amounts of experimental data encompassing all possible changes. For diagnosing SARS-CoV-2, this type of AI is critical for optimizing PCR tests because it enables rapid diagnosis and reduces false positives. Our method can also be used for other types of molecular analyses.

## 1. Introduction

Since the appearance of Severe Acute Respiratory Syndrome Coronavirus 2 (SARS-CoV-2) was first announced in China and the World Health Organization declared the COVID-19 outbreak a pandemic, the number of infected and dead cases has mounted. In countries with limited resources, the health systems are overloaded and the surveillance systems are deficient, allowing SARS-CoV-2 to spread rapidly. Therefore, developing new strategies for managing the pandemic are required [1] because effective medications, treatments, or vaccines have not been developed thus far. Moreover, there is uncertainty regarding the behavior of the COVID-19 pandemic, the increase in respiratory diseases, the potential for new pandemics to emerge, and the need to protect the population and economy.

Molecular and computational techniques, information and communication technologies, artificial intelligence (AI), and big data have advanced and can help manage the pandemic. AI has proven to be a very powerful tool in this pandemic. Chinese Baidu, Oregon State University, and the University of Rochester published a protein folding prediction algorithm, which is much faster in comparison with traditional algorithms in predicting the structure of the viral ribonucleic acid (RNA), and provides information on the spreading mechanism of viruses [2]. Studies predicting the structures of coronavirus proteins have been conducted during the pandemic [3,4,5]. Other fronts include an exchange of knowledge, the evolution of the pandemic, the support of healthcare personnel, and its application as a tool for population control [6,7,8,9].

Machine learning (ML) methods have been revolutionary in medical applications for the detection [10,11], diagnosis, and treatment of diseases such as cardiovascular diseases [12,13] and eye diseases, e.g., diabetic retinopathy [14] and corneal abnormalities [15,16].

Additionally, ML methods have been used for the detection and diagnosis of COVID-19. ML has been used for (a) predicting the 3D structure of proteins for designing drugs against COVID-19 [17,18], (b) analyzing chest CT scans autonomously in patients with COVID-19 [18], thereby allowing the detection of patterns generated by the virus in radiological results, because these analyses are labor-intensive for the radiologist, which can contribute to the generation of false negatives (FNs), (c) pandemic prediction for hospitalization, data from blood samples, and tomography to identify high-risk patients and predict whether they will develop respiratory difficulties, (d) studying the spatial–temporal dynamics of pandemics to understand how the virus is transmitted and spread, thus establishing quarantine standards from a social point of view, and (e) management of the COVID-19 literature, where ML methods help provide organized and classified data that can greatly accelerate the investigation of COVID-19. Despite the benefits of ML in effective patient care, limitations, including requirement of feature extraction and manual selection, poor performance for unbalanced datasets, overfitting, complexity, and time consumption, are present.

Given the immense need for AI in developing innovative solutions for this pandemic, the focus was placed on identifying the most promising methods for the detection and the rapid diagnosis of SARS-CoV-2 with high accuracy [19], although new strategies to improve COVID-19 diagnosis were also designed [20,21,22].

Studies, such as those carried out by Allam et al. (2020) [23], have reported that the use of deep learning methods (DLM), namely, convolutional neural network (CNN) and convolutional short-term long-term memory, has some advantages and improvements in COVID-19 detection accuracy compared to other ML methods.

Although the ML and DLM methods are a good option against COVID-19, there is a great limitation in “the reliable data”, whereby the availability of data hinders the realization of these models and the verification of their learning capacity. Moreover, most studies are aimed at helping doctors make an accurate diagnosis on the basis of pattern detection through COVID-19 images [23].

Therefore, we propose a new application based on AI. The aim of this study was to identify anomalies during PCR amplification for SARS-CoV-2 detection using AI. Our method can help overcome two problems: (1) deficiency in the rapid and timely diagnosis of SARS-CoV-2 by PCR and (2) shortage of trained molecular biologists in health institutions.

Globally, real-time PCR is considered the gold standard test for the diagnosis of SARS-CoV-2 [20,24,25]. The availability of commercial and in-house real-time reverse transcription PCR (RT-PCR) assays for detecting SARS-CoV-2 is limited. Moreover, the methods can vary greatly, and the processing capacity of the laboratories depends on the availability of inputs. Therefore, the development of technologies should be focused on making quick and reliable diagnoses.

The exponential increase in the number of PCR tests necessary to cover the enormous demand has caused a problem regarding analysis of PCR results, making one-to-one verification of the results cumbersome, and has led to reliance on closed-analysis software. For the real-time RT-PCR assays in use, analysis could use as much as four channels, generating four PCR curves for each patient sample. This corresponds to 46 features for each channel that must be analyzed by a laboratory specialist, which is consistent with our laboratory (Simon Bolivar University, Barranquilla) that analyzes more than 500 samples each day. Thus, we developed AI that detects anomalies and streamlines sample analysis for diagnosing COVID-19 to simplify the work of specialists. Our AI can recognize variations or noise that can affect a curve and can generate an alert for issues in the amplification process, using the curve patterns of the PCR graphs.

## 2. Results and Discussion

### 2.1. Principal Component Analysis

A principal component analysis (PCA) was conducted on real-time RT-PCR curves (RT-PCR-c) obtained from the Virology Laboratory of the Simon Bolivar University for the diagnosis of SARS-CoV-2 between April and May 2020 (all data). A total of 14230 RT-PCR-c, after 46 cycles of amplification, were analyzed. The first PC (PC 1) had 58% data variance. Two groups were found in association with the score on PC 1 and PC 2, as shown in Figure 1a. RT-PCR-c for each group can be visualized in Figure 1b. In the first group, different forms of RT-PCR-c were observed, whereas only one form (sigmoidal shape) was found in the second group.

Moreover, three classes were created, namely, correct amplification (+), no amplification (−), and abnormal amplification (Aa).

The + class is a corrected form of the RT-PCR-c amplification, which must describe an s-form or sigmoidal curve. RT-PCR-c must have a quantitation cycle (*Cq*) between 10 and 40 cycles for the diagnosis of SARS-CoV-2. *Cq* describes the cycle in which relative fluorescence is either detectable or exceeds a set threshold, according to the instrument’s signal-to-noise ratio (S/N).

An analysis of loading or weighting on the PC indicated that PC 1 was related to the + class (Figure 2a), due to which the loading on PC1 had the same form. The + class was found in the second group, which explains why the score in group 2 had values higher than that in group 1.

The − class included two options. The first was a corrected form of RT-PCR-c amplification, with an s-form, but with a *Cq* higher than 40 for SARS-CoV-2 diagnosis, whereas, the second included a null amplification, where the fluorescence does not exceed the threshold for *Cq*. This class can be found in groups 1 and 2. The Aa class is described by loading on PC 3 to PC 10 and for amplifications that do have sigmoidal shape. These characteristics could be due to sample contamination, electrical problems, use of PCR tubes from different manufacturers, etc. These anomalies can have a *Cq* value lower than 40 and could generate false positives (FPs) and must be categorized as invalid results.

### 2.2. The ML Model

A well-characterized portion (W-CP) for each class was extracted from all data using a visualization method of the curves one-to-one, but no data could be obtained from class ±. Therefore, an ML model based on the remaining three classes was generated. Different models of ML algorithms, namely, K-neighbor classifier, support vector machine for classification (SVC), decision tree classifier, and random forest classifier (RFC), were tested with 20% selected data for three classes. RFC was identified as the best model (number of trees in the forest = 1000 and minimum number of samples required to split an internal node = 2). The evaluation criteria (precision, recall, f1-score, and accuracy) and the confusion matrix calculated for the RFC model over the test data are listed in Table 1.

The model had good accuracy but could only be applied to these data if changes in data acquisition occurred. Moreover, it was necessary to redo the model. Thus, the model obtained conventionally was not viable (1) when *Cq* was used for the positive diagnosis of the amplification or a change in the case of a negative value, and (2) when the number of cycles used in the analysis was different compared to what the data were trained on (for example, in this study, 46 cycles were used, which generated 46 variables or characteristics when PCR analysis was performed); therefore, with a different number of cycles applied, the model may become unusable. Furthermore, if an instrument with better S/N is used, the threshold for the *Cq* creating FPs can be changed. Therefore, AI-based self-learning algorithm on simulated data was proposed (see Algorithms 1 and 2).

### 2.3. Data Simulation

The AI algorithm was created in association with simulated data from PCR amplification. To simulate data for classes + and −, a logistic function was used in Equation (1):(1)$$F\left(x\right)=\frac{A_{p}}{1+exp\left(-b*\left(x-Cm\right)\right)}+\frac{rand\left(1\right)*Thd}{3},$$
where *F* is the relative fluorescence, *x* is the number of cycles, Cm is the value in *x* when *F*(*x*) has the maximum rate of change or change of concavity, *A_p_* is the maximum amplitude, *b* is the growth rate, *Thd* is the threshold (*Thd*) for determining *Cq*, rand is a function that returns random numbers from 0 to 1, and the *Thd*∗*rand* multiplication simulates noise (Figure 3).

The classes were designed by changing the parameters and according to *Cq* ranges, as shown in Table 2. This parameter was changed randomly in the prescribed range. Instrumental noise, in addition to fluorescence measurement, for the instrument used for the tests was 6.7 units of relative fluorescence (RFU). This generated an S/N of 20 RFU. Using this value as *Thd*, it was possible to determine Cm, by clearing this from Equation (1) for *F* = *Thd* and *x* = *Cq* (Equation (2)).
(2)$$Cm=\frac{ln\left(\frac{A_{p}}{Thd}-1\right)}{b}+Cq.$$

Different ranges of *Cq* were used for each class, and random variations of b and *Ap* in the ranges specified are shown in Table 2, similar to *Cq*. One thousand curves were generated for each class. The *Ap* parameter ranged from 2–1000 × *Thd*, and the *b* parameter ranged from 0.2 to 1.0 (see Figure 3a, simulation algorithm).

Data simulation for Aa was generated from the loading normalized between 0 and 1 for PC 3 to PC 10 (*L_PC*), as well as the rand function that can be seen in Equation (3).
(3)$$F\left(x\right)=A_{p}\left(\frac{L\_PC\left(x\right)-L\_PC_{Min}}{L\_PC_{Max}-L\_PC_{Min}}\right)+rand\left(1\right)*Thd,$$
where *L_PC_Max_* and *L_PC_Min_* are the maximum and minimum values of *L_PC*, respectively (see Algorithms 1 and 2). The simulated data are shown in Figure 3.

This function returns random number between two numbers.
**Algorithm 1** Random Function (Matlab)function y=aleat(x,x2)    dang=abs(x-x2);    dt=rand() * dang;    if x > x2    y=x2+dt;    else    y=x+dt;    **end****end****Algorithm 2** Simulation Algorithm Using PC (Matlab)pos=zeros(1000,46); % class + neg=zeros(1000,46); % class – Aa=zeros(1000,46); % class Aa AaEx=PCA; % Principal component AaExn=zeros(20,46);k=1;**while** k < 21 % the # PC was 20     AaExn(k,:)=(AaEx(k,:)-min(AaEx(k,:)))./(max(AaEx(k,:))-min(AaEx(k,:)));     r=1;     **while** r < 51       one=ones(1,46);       s=1;        while s < 47        one(s)=one(s) * rand();        s=s+1;        end       *Ap*=aleat(140,300);       Apl=aleat(0,100);       Aa(50 * (k-1)+r,:)=(*Ap*. * AaExn(k,:))+one-Apl;       r=r+1;     **end**
     k=k+1;**end**
i=1;*Thd*=20; **while** i < 1001     b=aleat(0.02,0.5); % parameter b     Cqp=aleat(10,40); % *Cq* for +     Cqn=aleat(41,100); % *Cq* for –     *Ap*=aleat(40,2000); % parameter *Ap*     j=1;     Cmp=((log((*Ap*/*Thd*)-1))/b)+Cqp; % *Cq* for +     Cmn=((log((*Ap*/*Thd*)-1))/b)+Cqn; %*Cq* for –     while j < 47       pos(i,j)=(*Ap*./(1+exp(-b. * (j-Cmp))))+(6 * rand());       neg(i,j)=(*Ap*./(1+exp(-b. * (j-Cmn))))+(6 * rand());       j=j+1;     
**end**
     i=i+1;**end**X=[pos; neg; Aa];

Different ML methods were used to obtain a model based on simulated data. The models were tested in two ways: (1) for 20% of the simulated data, and (2) for the W-CP data. RFC was identified to be the better method using simulated data (S-Data-model). Table 3 shows the evaluation parameters.

### 2.4. Big Data Classification

All data were classified using the S-Data-model (Figure 4a). Subsequently, the data were visually revised by class, and a label was designated following the prediction of the S-Data-model, although false positivity and misdetection were reclassified on the basis of the findings obtained in the revision. An overlap created a large number of FPs and FNs between Aa and − and +.

To improve prediction, a new simulated data binary RFC model (SB-model_RFC) was generated using only two classes (number of trees in the forest = 800, minimum number of samples required to split an internal node = 2). The first included + alone; the second was a combination of − and Aa (−, Aa). Moreover, different preprocessing strategies were tested. The results that were obtained using all data as a test are shown in Table 4. The best preprocessing strategy was to apply the first derivative (D[·], the functions that are inside the brackets are preprocessed) followed by the smoothening of the data (Sn[·]) with 13 points and, subsequently, generate circular shifts (Cf[·]), where the maximum of the first derivative is centered on the data series. Finally, normalization (N[·]), subtracting the mean value and dividing it by the standard deviation, was performed (Figure 4b). Cf[·] refers to the circular shifts for the values in the vector, using shift size elements (SSEs). SSE is an integer scalar, where the *n*-th element specifies the shift that amounts to the *n*-th position of the vector. If an element in SSE was positive, the values of the vector shifted to the right. If an element in SSE was negative, the values of the vector shifted to the left. If an element in SSE was 0, the values in that dimension were kept in place. SSE was derived from the cycle number for the maximum of the first derivative, which was Cm for the + classes, and this value was subtracted from 23, which was put at the center of the cycle number to find the SSE (Figure 4b). Because the SB-model_RFC cannot differentiate between − and Aa, a second simulated data binary RFC model (SB2-model) trained with only data from − and Aa was generated. If the SB-model_RFC predicted that the amplification belonged to class −, Aa, then it was analyzed by the SB2-model (Table 4 and Figure 5a). However, a simpler strategy for differentiating between Aa and − included a simple decision algorithm (SD-A). If the fluorescence between cycle numbers 0 to 40 exceeded *Thd* or was less than *Thd*, it was classified as Aa or −, respectively. This strategy gave better results than the SB2-model, yielding an accuracy score of 1.00.

### 2.5. Challenges of the Methodology

Models are trained with simulated data rather than experimental data because experimental data must be characterized and these steps can be tedious for the analyst. Data simulation hastens the procedure and the model can be generated quickly. Models trained with simulated data may have an advantage because only certain experimental parameters need to be known. In this study, the simulation of real-time RT-PCR curve analysis for + and − from the parameters for the determination of specific detection (*Cq*) and the instrument signal to noise ratio (*Thd*) shows that it is possible to use this methodology in other PCR machines for detecting other targets by real-time RT-PCR. For this purpose, data simulation and model training can be realized using the parameters *Cq*, *b*, *Ap*, and *Thd*, which must be done for each PCR machine and type of target analysis. The user can input the *Cq*, *Ap*, *b*, and *Thd* values and the software generates the simulated data and the model. However, these models have limitations. The data simulated for Aa were generated from the PC of the specific data obtained from the PCR machines in our laboratory. Thus, different PCR machines will give different signals and PC values.

#### 2.5.1. Data Simulation from Random Function (DSRF)

To avoid the problems described in Section 2.4, simulated Aa data were generated from a random function, where a random value was generated ranging from *Thd* to *Ap*, ensuring that it exceeds *Thd* (see Algorithm 3). New models using different ML algorithms were generated and tested with all data (Table 5). In Table 5, a comparison between accuracy and log loss parameters from different ML methods is shown. Because linear discriminant analysis (LDA) presented the highest accuracy value of 97.6 and a low log loss of 0.1, it was the best method.
**Algorithm 3** Simulation Algorithm Using Random Function (Matlab)pos=zeros(1000,46); % class + neg=zeros(1000,46); % class – Aa=zeros(1000,46); % class Aa i=1;*Thd*=20;**while** i < 1001      b=aleat(0.02,0.5); % parameter b      Cqp=aleat(10,40); % *Cq* for +      Cqn=aleat(41,100); % *Cq* for –      *Ap*=aleat(40,2000); % parameter *Ap*      j=1;     Cmp=((log((*Ap*/*Thd*)-1))/b)+Cqp;     Cmn=((log((*Ap*/*Thd*)-1))/b)+Cqn;     **while** j < 47       pos(i,j)=(*Ap*./(1+exp(-b. * (j-Cmp))))+(*Thd* * rand()/3);       neg(i,j)=(*Ap*./(1+exp(-b. * (j-Cmn))))+(*Thd* * rand()/3);       if j<5       Aa(i,j)= aleat(*Thd*,*Ap*);       
**else**
       Aa(i,j)= aleat(10 * *Thd*,2 * *Ap*);       
**end**
       j=j+1;     
**end**
     % data smoothing      Aa(i,:)=(Aa(i,:)+((circshift(Aa(i,:)′,1)′))+((circshift(Aa(i,:)′,2)′))+((circshift(Aa(i,:)′,3)′)))./4;     Aa(i,:)=(Aa(i,:)+((circshift(Aa(i,:)′,1)′))+((circshift(Aa(i,:)′,2)′))+((circshift(Aa(i,:)′,3)′)))./4;     % offset referring to the initial intensity     Aa(i,:)=Aa(i,:)-mean(Aa(i,1:4));     i=i+1;**end**Xa=[pos; neg; Aa]

The evaluation criteria and confusion matrix parameters for the simulated data binary model, using the LDA classification algorithm (SB-model_ DSRF _LDA), are shown in Table 6. Considering previously described methodologies, two binary models were generated to improve prediction. The first binary model occurred between the groups + and −, Aa and the second model occurred between − and Aa. If we compare the accuracy values obtained in Table 4 (where the model was generated using simulated data, which uses experimental PC values for the simulation) with the accuracy values obtained in Table 6 (where the model was generated from simulated data, which are independent of experimental data), we can see that the results are very similar. Therefore, either method can be used, particularly the model generated by the random function, because it does not depend on experimental data.

#### 2.5.2. Data Simulation from ML (DSML)

Data simulation was performed using the kernel density estimator [26,27,28] (KDE) through GridSearchCV to determine the best parameter of bandwidth (BW). KDE is needed for generating data from an existing dataset [29,30]. The BWs tested were 0.01 to 1, with intervals of 0.05. Small portions of data of different classes (400 samples) were used for training the generator sample dataset. Six thousand RT-PCR-c data were generated and used for training different ML models. The models were tested with all data. A comparison between accuracy and log loss using DSML is shown in Table 5. LDA presented the highest accuracy value = 98.0 and a low log loss = 0.18 and was, thus, the best algorithm.

The evaluation criteria and confusion matrix parameters for the simulated data binary model using the LDA classification algorithm (SB-model_ DSML_LDA) are shown in Table 7. This model shows that data can be generated by using estimators based on ML methods. The values found are similar to those of other simulation strategies, which indicates that data generation can be assisted by ML methods. However, the difference is that a small sample of experimental data is necessary to generate the simulation, unlike the previous strategy, which does not use any sample. This strategy is more accurate if the PCR machine or methodology is changed because it includes variations, which earlier strategies did not.

### 2.6. Implementation of AI

AI was implemented using a web interface (WI). PCR curves exported from the Biorad CFX Maestro software of the PCR instrument were loaded onto the WI and configured, depending on the type of COVID-19 test. In our laboratory, two protocols were used. The first amplifies three SARS-CoV-2 genes, E, RdRp, and N in the FAM, Cal Red 610, and Quasar 670 channels, respectively, which are symbols of the fluorophores used for fluorescence emission when the gene is amplified and are indicative of SARS-CoV-2 RNA. A fourth internal control gene in the HEX channel is used to verify the nucleic acid extraction. To test positive for COVID-19, any of the first three genes must be amplified. Similarly, for a SARS-CoV-2 negative result, none of the first three genes must be amplified and amplification should occur only for the control gene (Figure 5b). If none of the genes are amplified, the test is considered invalid. Problems arise when the amplification is abnormal due to various factors discussed above, in which case the test must be invalidated. This is where AI contributes to the analysis as the PCR curves, otherwise reviewed by the analyst one-by-one, because relying on *Cq* alone can generate false results. AI makes work easier by predicting these anomalies. When the A curve is classified by the AI as Aa, a signal on the WI screen appears, and the analyst can review it by clicking on it and visualizing the curve, as shown in Figure 6. Thus, the analyst can decide to invalidate the result.

The second protocol is based on the assay by Corman et al. (2020) [31] and uses one fluorophore, FAM, for the detection of the SARS-COV-2 genes E and RdRp and the human RNase P gene as an internal control. As SARS-CoV is not circulating in Colombia, the PAHO recommends using only the E gene for epidemiological surveillance and diagnostics. Therefore, if the FAM channel for the E gene is amplified, it is considered positive for SARS-CoV-2. If FAM RNase P is amplified and FAM E is not amplified, the result is considered negative. Moreover, if neither is amplified, the result is invalid. Therefore, AI performs the same work as the first protocol in detecting the presence of Aa (Figure 5c).

## 3. Materials and Methods

### 3.1. Clinical Specimens

AI analysis was performed on the basis of 3559 real-time RT-PCR results (including controls). The clinical specimens, including nasopharyngeal swabs or bronchoalveolar lavage, were collected from 2390 patients suspected of having COVID-19. The processing of specimens was performed in a class II biological safety cabinet using biosafety level two (BSL2) facilities. Specimens were heat-inactivated at 70 °C in an oven and stored at –70 °C until testing.

### 3.2. Nucleic Acid Extraction

Nucleic acid was extracted from clinical specimens using the Quick-Viral RNA Kit (Zymo Research, Irvine, CA, USA) following the manufacturer’s instructions. Briefly, 200 μL of nasopharyngeal swab or aspirate was washed with the buffer and 75% ethanol and centrifuged at 12,000× *g* for 2 min through a column provided by the kit. The RNA was eluted with 20 μL of RNase-free water into nuclease-free vials and either tested immediately or stored at −70 °C.

### 3.3. PCR Method

The real-time RT-PCR assay was performed using the Real-Time kit Allplex 2019-nCoV Assay (Seegene, South Korea), according to the manufacturer’s instructions. The assay was based on the detection of a multiplex of three SARS-CoV-2 genes, E, N, and RdRp in the FAM, Quasar 670, and Cal Red 610 channels, respectively. The kit contains an exogenous RNA that functions as an internal control in the Hex channel. The reactions were carried out on the BioRad CFX96 Real-Time PCR Detection System. If two or three genes were detected (Ct value <40), the sample was considered positive for the presence of the virus. If either N or RdRp gave a Ct value <40, the sample was positive for SARS-CoV-2. However, if only the E gene was present, the sample was presumed positive, because SARS-CoV could be also detected by this set of primers and probe. A second real-time RT-PCR protocol adapted from Corman et al. (2020) [31] was also tested. Briefly, a 20 μL reaction was prepared containing 5 μL of RNA and 10 μL of 2× reaction buffer, provided with the iTaq universal Probe One-Step Kit (BioRad, Hercules, CA USA); primer and probe sequences used are as reported in Corman et al. (2020) [31]. All oligonucleotides were synthesized and provided by LGC Biosearch Technologies (Petaluma, CA USA). Thermal cycling was performed at 55 °C for 10 min for reverse transcription, followed by 95 °C for 3 min and 45 cycles of 95 °C for 15 s and 58 °C for 30 s. A Ct <40 for the E gene was considered positive for the presence of SARS-CoV-2 RNA.

### 3.4. ML Methods Analysis and Data Simulation

The ML methods were acquired to build classification models using the library of sklearn 3.2 in Python 3. The package of libraries in scikit-learn [29,32] provided a set of open-source software, including efficient AI techniques for the Python programming language. The different algorithms of ML used included KNC, SVC, the decision tree classifier, RFC, quadratic discriminant analysis (QDA), and LDA. Some of these methods use default parameters, which were optimized to obtain the highest accuracy.

The performance of the models was evaluated using recall, precision, f1-score, support accuracy, and log loss. In a binary model, the recall for positive and negative cases is known as sensitivity and specificity, respectively. Precision involves the ratio between TP and (TP + FP), where TP is the number of true positives and FP is the number of FPs. The f1-score is also known as the balanced f-score. The f1-score is an indication of a weighted average of precision and recall, with a best value of 1. The formula for the f1-score is shown in Equation (4).
(4)$$f1-score=2*\frac{precision\;*\;recall}{precision\;+\;recall}$$

Log loss is an important classification metric based on probabilities. This is a metric for record loss. Log loss is a good metric for comparing models. A lower log loss value means better prediction.

Data simulation was performed using the MATLAB software. The algorithm is presented in the Appendix A

### 3.5. Web Platform Design for the Implementation of AI

The WI was designed in HTM5, node JavaScript, and CSS. In addition, it was fed with the PCR curves exported from the Biorad CFX Maestro software of the PCR instrument. The platform organized, read, visualized, and connected with the application programming interface (API) loading of the AI model. Subsequently, the platform created an interactive environment where the Aa ratings from the AI were displayed as flickering.

## 4. Discussion and Conclusions

The SARS-CoV-2 pandemic has increased the requirement of PCR tests for diagnosis. Therefore, alternative diagnostic methods are necessary to accelerate diagnosis, especially in countries where the number of diagnostic laboratories is limited.

Our study adds value to the available state-of-the-art techniques. We developed an AI-based ML algorithm which use minimum available information in association with a set of data obtained under the scope of the study through the described simulations. This concept opens other perspectives for finding solutions quickly and in real time. In this application, it is possible to create an ML model where little experimental information is available and a quick response is required. Changes in data when conditions change, such as variations in PCR machines and amplification kits, require the generation of new models using experimental data that include these changes. In this study, we proposed that the changes can be included for generating simulated data and the challenge of including experimental data containing all possible changes can be avoided. Three simulation strategies were proposed; the first uses PCA as guiding information for simulating the training data, the second uses algorithms of random functions and the profile of amplification curves previously analyzed for generating simulated data, and the third involves simulated data generated using KDE algorithms from few experimental samples of each class. In conclusion, all strategies provided good results when validated with the experimental data. However, the second strategy is the most extreme as it requires less experimental information. This strategy could fail in other experimental systems but shows that models can be generated using this methodology. For that reason, the third strategy was planted where data simulation was done using small groups of experimental data that do not require a long acquisition time. Lastly, this type of AI for diagnosing SARS-CoV-2 is crucial for evaluating PCR tests, because it reduces human intervention in laboratory practice. Therefore, it has been implemented in our laboratory (Simón Bolivar University, Barranquilla). However, implementing our AI in other laboratories for validating our models, corroborating the results in the real world, and using these methods for other types of molecular analyses is warranted.

## Figures and Tables

**Figure 1 molecules-26-00020-f001:**
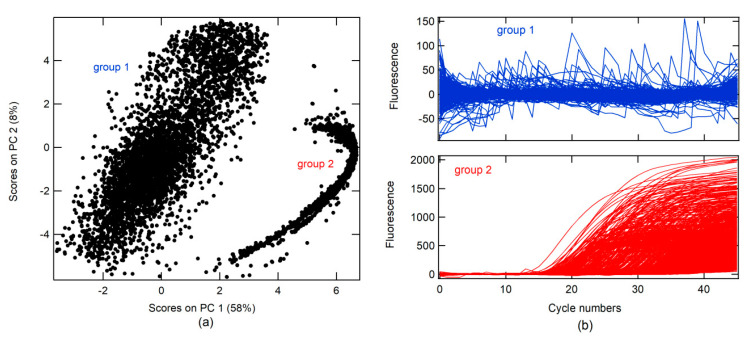
(**a**) Score plots of principal component (PC) 2 vs. PC 1 for real-time RT-PCR curves for SARS-CoV-2 diagnostics; (**b**) real-time RT-PCR curve plot for the two groups found during principal component analysis (PCA).

**Figure 2 molecules-26-00020-f002:**
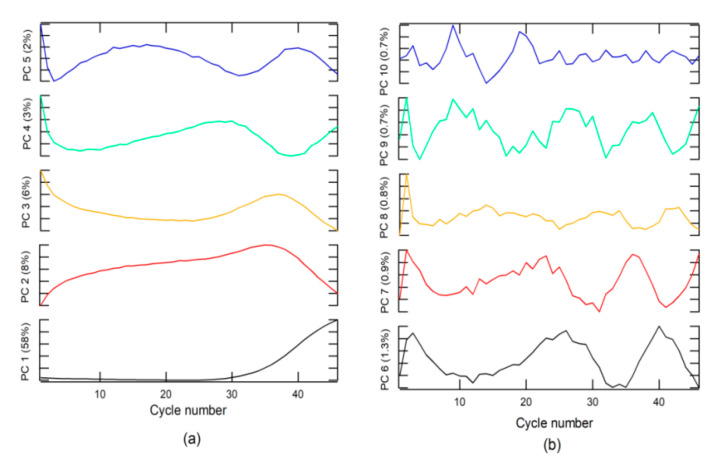
Loading plots for (**a**) principal component (PC) 1 to PC 5, and (**b**) PC 6 to PC 10.

**Figure 3 molecules-26-00020-f003:**
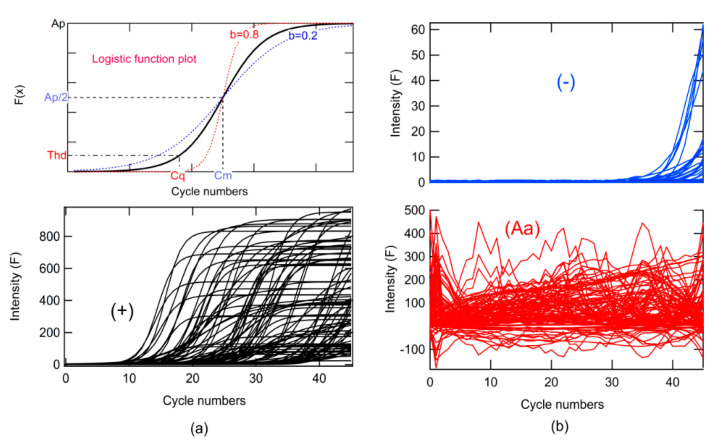
(**a**) Logistic function plot for different growth rate parameters used in the simulations and simulated real-time RT-PCR curves for a class; (**b**) simulated real-time RT-PCR curves for classes no amplification (–) and abnormal amplification (*Aa*).

**Figure 4 molecules-26-00020-f004:**
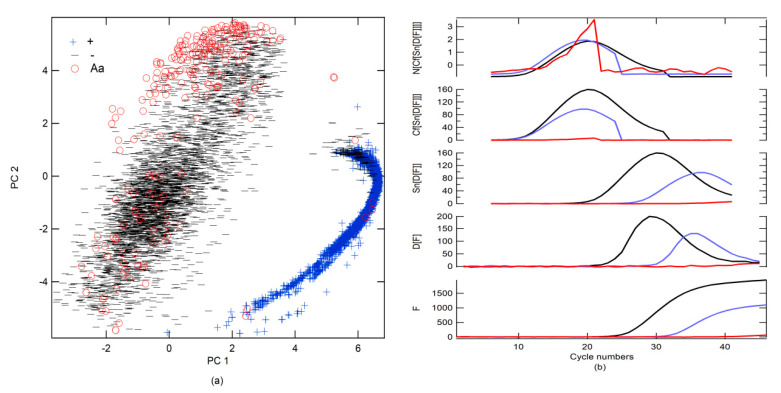
(**a**) Score plots of PC 2 vs. PC 1 for real-time RT-PCR curves for the diagnosis of SARS-CoV-2 and their classification based on the S-Data-model; (**b**) scheme of the best preprocessing combination, preprocessing sequence: N[Cf[Sn[D[F]]]].

**Figure 5 molecules-26-00020-f005:**
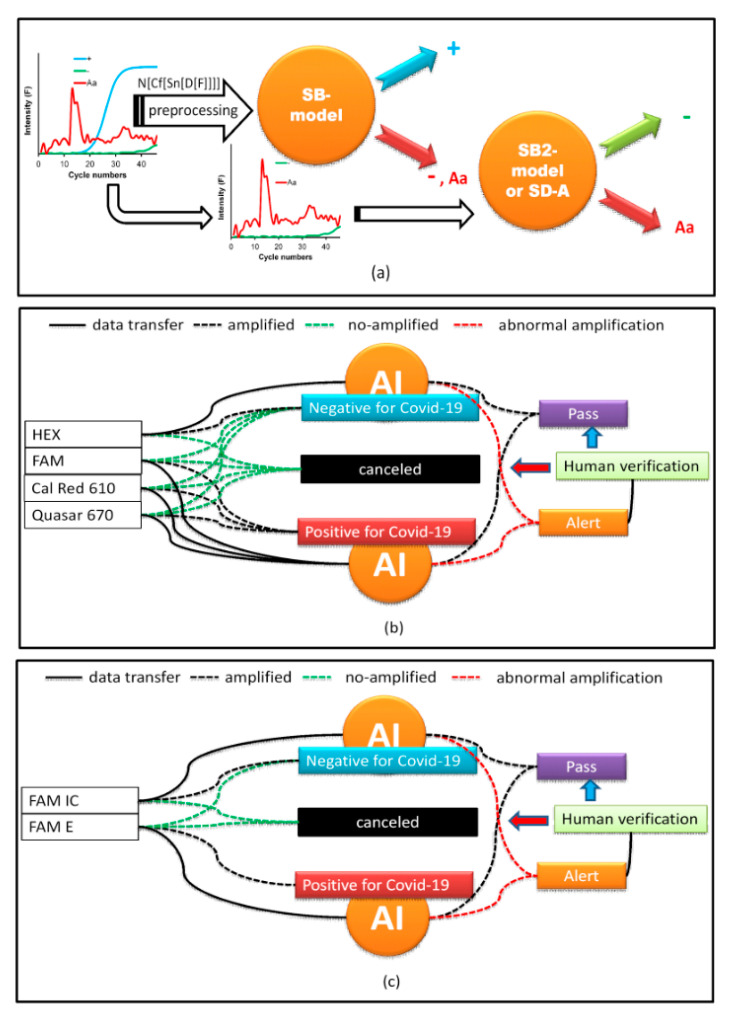
(**a**) The artificial intelligence (AI) classification scheme; (**b**) implementation of AI for the first PCR kit for COVID; (**c**) implementation of AI for the second PCR kit for COVID.

**Figure 6 molecules-26-00020-f006:**
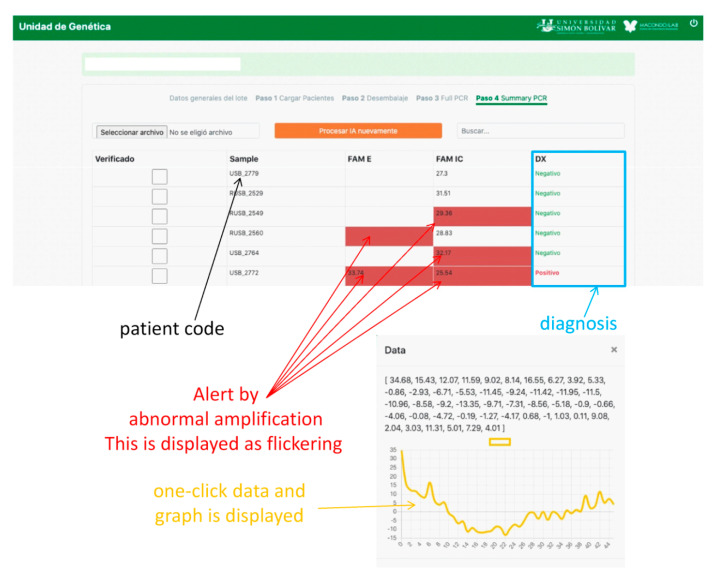
Web platform design for the implementation of AI.

**Table 1 molecules-26-00020-t001:** Confusion matrix and evaluation criteria for the random forest classifier (RFC) model for the well-characterized portion (W-CP).

Test for 20% of W-CP								
	Precision	Recall	f1-Score	Support	Accuracy	Matrix of Confusion		
	Model					+	−	?
+	0.976	1.000	0.988	40	0.972	40	0	0
−	0.961	1.000	0.980	49		0	49	0
Aa	1.000	0.700	0.824	17		1	2	14

**Table 2 molecules-26-00020-t002:** Parameters for logistic functions of classes + and −.

	+	−
*Cq*	10 to 40	40 to 60
*b*	0.2 to 1.0	0.2 to 0.8
*Ap*	40 to 1000	40 to 1000

+, −, abnormal amplification (Aa), growth rate (b), quantitation cycle (*Cq*).

**Table 3 molecules-26-00020-t003:** Confusion matrix and evaluation criteria for the random forest classifier (RFC) model of the S-Data-model.

Test for 20% of Simulated Data								
	Precision	Recall	f1-Score	Support	Accuracy	Matrix of Confusion		
	Model					+	−	Aa
+	1.000	0.989	0.995	93	0.952	92	1	0
−	0.989	1.000	0.994	87		0	87	0
Aa	0,873	1.000	0.932	117		0	0	117
Test for W-CP								
+	0.915	0.993	0.953	152	0,960	151	0	0
−	0.984	0.996	0.990	255		1	254	0
Aa	1.000	0.859	0.924	142		13	4	122

**Table 4 molecules-26-00020-t004:** Confusion matrix and evaluation criteria for the test using all data for SB-model_RFC and SB2-model.

Test of All Data for the SB-Model_RFC							
	Precision	Recall	f1-Score	Support	Accuracy	Matrix of Confusion	
	Model					+	−, Aa
+	0.955	0.979	0.967	5938	0.972	5811	127
−, Aa	0.984	0.967	0.976	8284		272	8012
**Test of—and Aa of All Data for the SB2-Model**							
	**Precision**	**Recall**	**f1-Score**	**Support**	**Accuracy**	**Matrix of Confusion**	
	Model					−	Aa
−	0.990	0.950	0.970	7287	0.948	6923	364
Aa	0.718	0.930	0.810	997		70	927

simulated data binary RFC model (SB-model_RFC) and second simulated data binary RFC model (SB2-model).

**Table 5 molecules-26-00020-t005:** Test comparison for different machine learning methods for different training strategies.

	DSRF		DSML	
Methods	Accuracy	Log Loss	Accuracy	Log Loss
KNC	97.5	0.6	93.0	1.2
SVM	96.6	0.1	97.4	0.14
RFC	92.2	0.2	96.1	0.2
QDA	85.5	3.9	94.3	1.3
LDA	97.6	0.1	98.0	0.18

**Table 6 molecules-26-00020-t006:** Confusion matrix and evaluation criteria for the test using all data for the SB-model_ DSRF _LDA and the SD-A.

Test of All Data for SB-Model_ DSRF _LDA (Preprocessing = N[Cf[Sn[D[F]]]])							
	Precision	Recall	f-Score	Support	Accuracy	Matrix of Confusion	
	Model					+	−, Aa
+	0.970	0.971	0.971	5801	0.976	5635	166
−, Aa	0.980	0.979	0.980	8287		173	8114
**Test of—and Aa of All Data for SD-A**							
	**Precision**	**Recall**	**f-Score**	**Support**	**Accuracy**	**Matrix of Confusion**	
	Model					−	Aa
−	1.000	1.000	1.000	7289	1.000	7289	0
Aa	1.000	1.000	1.000	998		0	998

simulated data binary model, using the LDA classification algorithm (SB-model_DSRF _LDA), simple decision algorithm (SD-A) and preprocessing sequence: N[Cf[Sn[D[F]]]].

**Table 7 molecules-26-00020-t007:** Confusion matrix and evaluation criteria for the test using all data for the SB-model_ DSML_LDA and the SD-A.

Test of All Data for SB-Model_ DSML_LDA (Preprocessing = N[Cf[Sn[D[F]]]])							
	Precision	Recall	f-Score	Support	Accuracy	matrix of Confusion	
	Model					+	−, Aa
+	0.969	0.970	0.970	5790	0.98	5616	174
−, Aa	0.979	0.979	0.979	8306		178	8128
**Test of—and Aa of All Data for SD-A**							
	**Precision**	**Recall**	**f-Score**	**Support**	**Accuracy**	**Matrix of Confusion**	
	Model					−	Aa
−	1.000	1.000	1.000	7289	1.000	7289	0
Aa	1.000	1.000	1.000	998		0	998

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
