# Peer review of "Anomaly Identification during Polymerase Chain Reaction for Detecting SARS-CoV-2 Using Artificial Intelligence Trained from Simulated Data"

_molecules, 2020, doi:10.3390/molecules26010020_

Round 1
Reviewer 1 Report
The paper addresses the topic of using artificial intelligence schemes in the scope of the SARS-CoV-2. The paper is interesting and focus on an important and timely topic. The reviewer thinks that the paper needs improvements before publication:
- The long abstract is not assertive. A more direct and shorter abstract would be preferable.
- On the other hand, The introduction to the topic and the review of the literature seem too short. Consider extending considerably
- The core of the paper is interesting, but readers have the impression that it is limited. Consider expanding
- Som minor detail. In fig 6 the graphical objects are to small. redesign the graphical layout in the figure
Author Response
The paper addresses the topic of using artificial intelligence schemes in the scope of the SARS-CoV-2. The paper is interesting and focus on an important and timely topic. The reviewer thinks that the paper needs improvements before publication:
Point 1: The long abstract is not assertive. A more direct and shorter abstract would be preferable.
Done
The abstract was reduced to 394 words
Point 2: On the other hand,The introduction to the topic and the review of the literature seem too short. Consider extending considerably,
Done
Point 3: The core of the paper is interesting, but readers have the impression that it is limited. Consider expanding
Done
Point 4: Som minor detail. In fig 6 the graphical objects are to small. redesign the graphical layout in the figure
Done

Reviewer 2 Report
In this manuscript the authors provide a mechanism to simulate PCR cycles for classification of normal or abnormal PCR amplification results. The simulation method itself is interesting, but I still have questions or comments regarding to the whole manuscript. Below please find my review.
- I can feel that the authors are very eager to call their methods “artificial intelligence.” Pardon me, but I do not see anything “AI” regarding to the simulation process. If I understand the method correctly, the authors basically designed a function with some randomization based on the extracted principle components. According to the definition of machine learning, which can be defined as the core of artificial intelligence, the machine need to “learn” from its results. There is nothing AI or ML in the simulation process, and the evaluation process made by several ML algorithms has nothing to do with the simulation. Please consider rename the title and the procedure itself.
- Continue from 1. The authors also claimed that their “AI” methodology can deal with the situation for different PCR machine or different cycle count (line 142-147). Seriously I do not see how this is possible for their methodology. The cycle count cannot be adjusted since their formula need the value of rate of change for each cycle count, and that different machine are going to generate different signals with different PC values or rate of changes for sure. Please remove these empty promises, and this needs to be done together with the revision and rename of “AI” for this methodology.
- There is an apparent gap between the real data and the simulation, in which the authors mentioned that no data could be obtained from class +- (line 130) while they can simulation +- data class (line 152). I do not understand how something that cannot be obtained can be simulated.
- rand(x) should be re-written as rand(1) for both formula 1 and 3 since you are returning random numbers from 0 to 1.
- What does it mean by “Cm is the value with the maximum rate of change or change of concavity” (line 153)? Since the formula states that “x - Cm", it should be a number or value related to x; Otherwise the formula does not make sense. Please clarify what Cm really is.
- What are the parameters for the machine learning algorithms? For example, what is the K for KNN? What is the kernel function for SVM? How many layers and what is the splitting function for decision tree? How many trees are there in the random forest? There are all very serious questions for machine learning, and I urge the authors to understand the impact of the parameters to the machine learning algorithms.
- Please release the data for examination purpose.
- citation 18-20 is very different from the claimed “AI.” To me citation 15-17 is also quite different. Please provide reason why the authors think they are the same or remove the sentences/citations.
Author Response
The response is attached to the file.
Thanks for your review, they were a lot of help to improve the paper

Round 2
Reviewer 1 Report
The reverse paper was improved and can be accepted for publishing. The reviewer suggests avoiding the use of the symbol "*" for multiplication since it makes confusion with the mathematical operation of convolution.
Author Response
English language and style was checked
Reviewer 2 Report
In general the authors tried to address my questions by focusing on the revision of titles and methodologies. I still have a few questions for the authors.
- The authors released the simulation code, which indicated that the crucial parameters for some unknown machine were indeed randomly generated. This was also written in the article (section 2.4). My first question, as I also pointed out in my original review, is why some randomly generated numbers can represent unknown/unidentified PCR machine models. It is understandable that different PCR machine are going to have different parameters; however I frankly do not think those parameters can be “simulated” by random functions. I urge the authors to check or show that their random function really works, perhaps by comparing among different PCR machines.
- Since the entire paper is about simulation, my second question is why the algorithm was embedded in supplementary materials instead of in the manuscript. Since the algorithm itself is critical to the success of simulation, I have a strong feeling that it should be in the paper. This of course needs to be coupled with the previous question that the random function can indeed serve their roles in the simulation process.
- Perhaps the authors were revising this articles in a hurry, the English wordings in the newly-added section is quite poor. Please revise the article on this aspect or perhaps ask an English editor to help.
- In the conclusion the authors again stated that “AI based on minimum information,” which has been revised elsewhere. Again, if you want to claim AI, show me specifically where the AI is. As I stated in my previous review, the simulation has nothing to do with AI/ML whatsoever. Please check my previous review and go through the entire article, including keywords, to make sure you do not make empty promises.
Author Response
1. The authors released the simulation code, which indicated that the crucial parameters for some unknown machine were indeed randomly generated. This was also written in the article (section 2.4). My first question, as I also pointed out in my original review, is why some randomly generated numbers can represent unknown/unidentified PCR machine models. It is understandable that different PCR machine are going to have different parameters; however, I frankly do not think those parameters can be “simulated” by random functions. I urge the authors to check or show that their random function really works, perhaps by comparing among different PCR machines.
When you say that the parameters are simulated by random functions, it is not entirely true; the parameters are randomly varied in a range where they are expected to be experimental.
This was modified
The classes were designed by changing the parameters and according to Cq ranges, as shown in Table 2. This parameter was changed randomly in the prescribed range
Different ranges of Cq were used for each class, and random variations of b and Ap in the ranges specified, are shown in Table 2, similar to Cq
2. Since the entire paper is about simulation, my second question is why the algorithm was embedded in supplementary materials instead of in the manuscript. Since the algorithm itself is critical to the success of simulation, I have a strong feeling that it should be in the paper. This of course needs to be coupled with the previous question that the random function can indeed serve their roles in the simulation process.
This was done. The algorithm is included
3. Perhaps the authors were revising this articles in a hurry, the English wordings in the newly-added section is quite poor. Please revise the article on this aspect or perhaps ask an English editor to help.
This was done
4. In the conclusion, the authors again stated that “AI based on minimum information,” which has been revised elsewhere. Again, if you want to claim AI, show me specifically where the AI is. As I stated in my previous review, the simulation has nothing to do with AI/ML whatsoever. Please check my previous review and go through the entire article, including keywords, to make sure you do not make empty promises.
The conclusion was revised